# A Coarse-to-Fine 3D U-Net Network for Semantic Segmentation of Kidney CT Scans

Yasmeen George

School of Information Technology
Monash University
yasmeen.george@monash.edu

**Abstract.** The number of kidney cancer patients is increasing each year. Computed Tomography (CT) scans of the kidneys are useful to assess tumors and study tumor morphology. Semantic segmentation techniques enable the identification of kidney and surrounding anatomy on the pixel level. This allows clinicians to provide accurate treatment plans and improve efficiency. The large size of CT volumes poses challenges for deep segmentation methods as it cannot be accommodated on a single GPU in its original resolution. Downsampling CT scans influences the segmentation performance. In this paper, we present a coarse-to-fine cascaded network based on 3D U-Net architecture for semantic segmentation of kidney CT volumes into three classes kidney, tumor, and cyst. A two stage approach is implemented where a 3D U-Net model is first trained on downsampled CT volumes to delineate kidney region. This is followed by another 3D U-Net model which is trained using using the full resolution images cropped around the areas of interest generated by first stage segmentation results. A set of 300 CT scans were used for training and evaluation. The proposed approach scored 0.9748, 0.8813, 0.8710 average dice for kidney, tumor and cyst using 3D cascade U-Net model. The performance of the cascade network outperformed other trained U-Net models based on 2D, 3D low resolution and 3D full resolution.

**Keywords:** Semantic segmentation· cascaded network · 3D U-Net · medical image diagnostics

## 1 Introduction

The number new cases with kidney tumors is increasing each year [2]. Globally, kidney cancer is the sixth most commonly diagnosed cancer for men and the ninth for women [1]. In Australia, it is the seventh most common cancer [2]. Kidney tumors can be classified as benign, indolent or malignant. Benign tumors can grow slowly but does not spread to other tissues. A malignant tumor is cancerous with renal cell carcinoma is the most common type of kidney cancer leading to 140,000 deaths annually worldwide [4]. An indolent tumor is also cancerous, but this type of tumor rarely spreads to other parts of the body. A great research effort is being invested on studying the relationship between

tumor morphology (size, shape and appearance) and surgical outcomes. Small tumors are often detected incidentally when the patient has a scan for an unrelated problem [3]. There is a need for automated semantic segmentation and classification methods to objectively quantify the severity of kidney tumors in order to better inform treatment decisions. This will also help doctors to solve diagnostic problems and improve efficiency.

Recent advances in imaging tools facilitate the detection and diagnosis of kidney tumors and contribute to preventive treatment of kidney cancer. Clinicians predominantly rely on imaging tests, primarily Computed Tomography (CT), to both diagnose and stage renal cell carcinomas [2]. CT uses x-rays to provide cross-sectional images of the body from different angles. The 2D slices are combined to form the final 3D volume for the kidney. CT scan reveals any abnormalities or tumors and can be used to measure the size of the tumor.

Semantic segmentation plays an important role in diagnostics support systems in the medical domain. It enables the identification of different objects in images on the pixels level. In this paper, we apply semantic segmentation for kidney tumor diagnosis where each voxel in the CT scan is labeled as background, kidney, renal tumors or cyst.

## 2    Methods

Inspired by nnU-Net work [3], we implemented a coarse-to-fine cascaded U-Net approach which has two stages. In the first stage, a 3D U-Net model is trained on downsampled images to roughly delineate kidney region. In the second stage, a 3D U-Net model is trained to have more detailed segmentation of the three classes (kidney, tumor, cyst) using the full resolution images guided by the first stage segmentation maps.

### 2.1    Training and Validation Data

In this paper, we use the kits21 dataset [1]. The KiTS21 challenge organizers have produced ground truth semantic segmentations for arterial phase abdominal clinical CT scans for training and validation. The dataset consists of 300 unique kidney cancer patients who underwent partial or radical nephrectomy for suspected renal malignancy between 2010 and 2020 at either an M Health Fairview or Cleveland Clinic medical center. Each CT volume consists of 29-1059 slices of 512-796×512 pixels. The voxel dimensions are [0.44-1.04, 0.44-1.04, 0.5-5] mm.

Regions in CT scan are for 3 different classes, kidney, tumor and cyst. The ground truth segmentation mask is provided for each class separately. A group of trainees that consists of medical students, undergraduates planning to study medicine, and one Computer Science PhD student annotate region of interests by placing 3D bounding boxes as well as guidance pins for axial slices. A group of experts that include radiologists and urologic cancer surgeons review the annotations and leave comments if necessary. The round review is repeated until

experts approve the annotation for accuracy and completeness. Finally the guidance for each region is sent individually to three laypeople for contour annotations. Trainees review the contour annotations and ensure that they adhere to the expert-approved guidance. Finally, contour annotations are postprocessed to generate segmentations.

All DL models in this paper used majority aggregation based ground truth segmentation for training and validation.

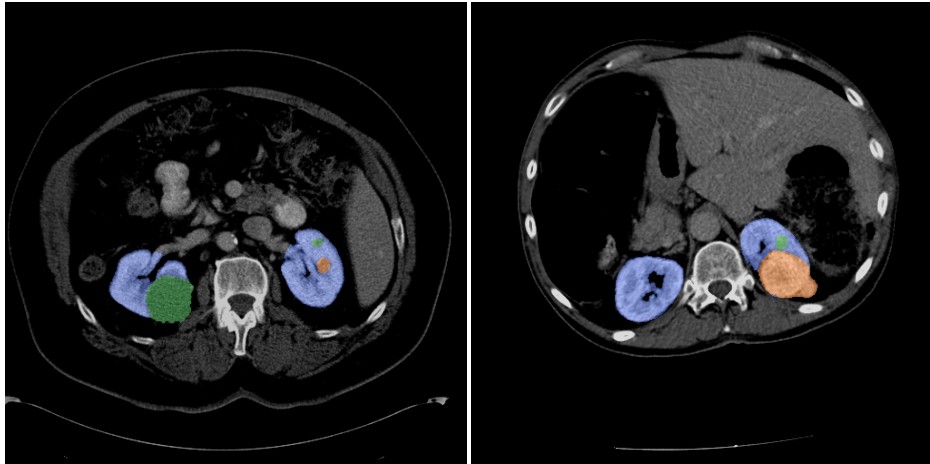

**Fig. 1.** Sample 2D slices from 2 CT scans with annotated kidney, tumor and cyst in blue, orange and green in order

## 2.2    Data Preprocessing

The CT intensities (HU) were transformed by subtracting mean and dividing by standard deviation. The data augmentation methods include random rotations, gamma transformation, and random cropping.

## 2.3    Proposed Method

The proposed approach is based on two stage cascaded network for kidney, tumor and cyst segmentation using 3D U-Net architecture. In the first stage, each CT scan was resampled using third order spline interpolation to a spacing of $1.99 \times 1.99 \times 1.99$ mm resulting in median volume dimensions of $207 \times 201 \times 201$ voxels. While in the second stage, a spacing of $0.78 \times 0.78 \times 0.78$ mm was used with median volume dimensions of $528 \times 512 \times 512$ voxels.

The 3D U-Net architecture had an encoder and a decoder path each with five resolution steps. The encoder part was performed using strided convolutions starting with 30 feature maps then doubling up each level to a maximum of 320.

The decoder part was based on transposed convolutions. Each layer consists 3D convolution with $3 \times 3 \times 3$ kernel and strides of 1 in each dimension, leaky ReLU activations, and instance normalization.

All the models were trained from scratch using 5-fold cross-validation with a patch size of $128 \times 128 \times 128$ that was randomly sampled from the input resampled volumes. The models were trained using stochastic gradient descent (SGD) optimizer for 1000 epochs using a batch of size 2 with 250 batches per epoch. The training objective was to minimize the sum of cross-entropy and dice loss.

## 3    Results

The proposed models are implemented using nnU-Net framework [3] with Python 3.6 and PyTorch framework on NVIDIA Tesla V100 GPUs. For performance evaluation, we report the average 5-fold dice coefficient and Surface Dice (SD). To compare the performance of the proposed network, we report the performance measures for 2d U-net model, 3D full resolution and 3D low resolution U-Net models. Table 1 displays the dice and SD scores for all trained models. The table shows that the dice scores of 0.9748, 0.8813, 0.8710 for kidney, tumor and cyst in order. We also visualise the segmentation results for all trained models in Figure 3.

**Table 1.** Average 5-fold performance measures for the trained U-Net models for kidney tumor segmentation

| U-Net model | Dice Scores | | | SD Scores | | |
|---|---|---|---|---|---|---|
| | Kidney | Cyst | Tumor | Kidney | Cyst | Tumor |
| 2D | 0.9615 | 0.7686 | 0.7367 | 0.9072 | 0.6154 | 0.5787 |
| 3D FullRes | 0.9691 | 0.8693 | 0.8535 | 0.9359 | 0.7626 | 0.7415 |
| 3D LowRes | 0.9685 | 0.8705 | 0.8568 | 0.9272 | 0.7508 | 0.7377 |
| 3D Cascade | 0.9748 | 0.8813 | 0.8710 | 0.9448 | 0.7728 | 0.7605 |

## 4    Discussion and Conclusion

The paper presents a cascaded deep neural network for semantic segmentation of kidneys and surrounding anatomy. The approach is based on 3D U-Net with two stages for training. The first stage model is trained on downsampled volumes while the second stage model is trained on the cropped region of interest in its full resolution. The models were able to accurately segment the kidney and tumor while the performance of cyst segmentation is lowest due to the small number of annotated cases with cyst.

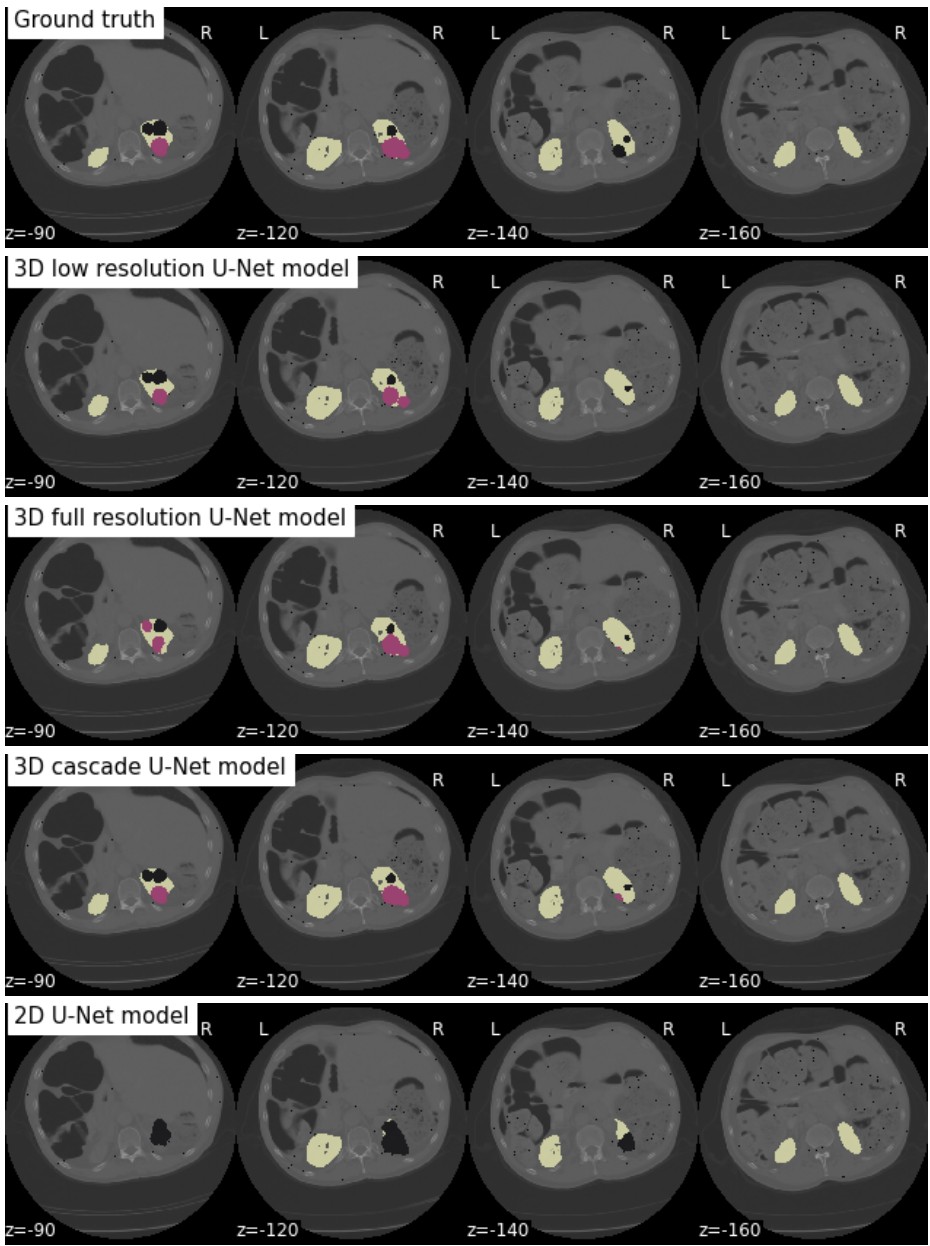

**Fig. 2.** Segmentation results for kidney, tumor and cyst using U-Net trained models

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
