# OpenReview forum: "A Coarse-to-Fine 3D U-Net Network for Semantic Segmentation of Kidney CT Scans"
_MICCAI.org/2021/Challenge/KiTS — Submitted to KiTS21 Challenge_

### Official Review · Reviewer_Lncj · 2021-08-30

**Rating:** 6

**Review:**

The authors present a coarse-to-fine approach with generally an adequate level of detail. It would be helpful if they could include one or more figures to summarize their approach. They should also be sure to include the final results once they are known. Perhaps these numbers would be better summarized in a table in the results section.

---

### Official Review · Reviewer_pkT3 · 2021-08-30

**Rating:** 6

**Review:**

### Overall

- In the abstract "This allows clinician" -> "This allows clinicians"
- An institutional email address for the corresponding author is preferred over gmail if possible

### Introduction

- "Benign tumor can ..." -> "Benign tumors can"

### Methods

- It would be nice if you could add a flow-chart to serve as a visual summary of your model. These are especially helpful for coarse-to-fine methods such as yours
- What method did you use for resampling?
- Which segmentations did you use for training? Majority voting to combine the multiple annotations, or some other approach?

### Results

- It would be nice if you could include a figure showing some examples of your predictions vs ground truth
- Please be sure to add final values here once the official results are known
- Are the results here on a per-batch or per-case basis?

### Discussion and Conclusion

- It might be nice to reiterate your results here

---

### Decision · Program_Chairs · 2021-08-30

**Decision:**

Major Revisions

**Comment:**

Please address the reviewer comments and resubmit